# Electrophilic Aldehyde 4-Hydroxy-2-Nonenal Mediated Signaling and Mitochondrial Dysfunction

**DOI:** 10.3390/biom12111555

**Published:** 2022-10-25

**Authors:** Sudha Sharma, Papori Sharma, Tara Bailey, Susmita Bhattarai, Utsab Subedi, Chloe Miller, Hosne Ara, Srivatsan Kidambi, Hong Sun, Manikandan Panchatcharam, Sumitra Miriyala

**Affiliations:** 1Department of Cellular Biology and Anatomy, Louisiana State University Health Sciences Center, Shreveport, LA 71103, USA; 2Department of Chemical & Biomolecular Engineering, University of Nebraska, Lincoln, NB 68588, USA

**Keywords:** oxidative stress, lipid peroxidation, 4-hydroxynonenal, mitochondrial dysfunction

## Abstract

Reactive oxygen species (ROS), a by-product of aerobic life, are highly reactive molecules with unpaired electrons. The excess of ROS leads to oxidative stress, instigating the peroxidation of polyunsaturated fatty acids (PUFA) in the lipid membrane through a free radical chain reaction and the formation of the most bioactive aldehyde, known as 4-hydroxynonenal (4-HNE). 4-HNE functions as a signaling molecule and toxic product and acts mainly by forming covalent adducts with nucleophilic functional groups in proteins, nucleic acids, and lipids. The mitochondria have been implicated as a site for 4-HNE generation and adduction. Several studies clarified how 4-HNE affects the mitochondria’s functions, including bioenergetics, calcium homeostasis, and mitochondrial dynamics. Our research group has shown that 4-HNE activates mitochondria apoptosis-inducing factor (AIFM2) translocation and facilitates apoptosis in mice and human heart tissue during anti-cancer treatment. Recently, we demonstrated that a deficiency of *SOD*2 in the conditional-specific cardiac knockout mouse increases ROS, and subsequent production of 4-HNE inside mitochondria leads to the adduction of several mitochondrial respiratory chain complex proteins. Moreover, we highlighted the physiological functions of HNE and discussed their relevance in human pathophysiology and current discoveries concerning 4-HNE effects on mitochondria.

## 1. Introduction

Reactive oxygen species (ROS) such as superoxide, hydrogen peroxide, hydroxyl radical, singlet oxygen, and lipid peroxyl radicals are natural by-products of cellular respiration, prostaglandin synthesis, and the cytochrome P450 system [1,2,3]. However, when the levels of these oxidants surpass endogenous levels, it leads to oxidative stress, producing cytotoxic effects and cell death [2]. Increased levels of ROS are seen in many diseases, as well as in severe tissue damage [4]. These include, but are not limited to, traumatic brain injury [5], Alzheimer’s disease, diabetes, aging [2], infertility [6], doxorubicin-induced cardiac injury [7], and cardiac ischemia-reperfusion injury [8]. Data from certain disease models have implied that mitochondrial production of superoxide radicals (O_2_^−^) is a major instigator of oxidative stress [9,10,11,12]. This oxidative stress leads to peroxidation in the lipid membrane and subsequent reactive aldehyde formation such as the formation of 4-HNE [2,6] (Figure 1). This highly reactive and cytotoxic aldehyde is then able to form covalent adducts with proteins, nucleic acids, and other lipids [2]. Adduction of 4-HNE to DNA leads to DNA damage, fragmentation, and subsequent cell death [7]. 4-HNE adducts with the protein via Schiff’s base formation and Michael addition by reacting with the aldehyde group and double bond in 4-HNE, respectively. These 4-HNE–protein adducts can lead to protein inactivation, aggregation, and change in protein conformation and function [13].

## 2. Generation of 4-HNE inside Mitochondria

When lipid membranes containing ω-6 polyunsaturated fatty acyl chains are subjected to ROS, one of the reaction products is 4-HNE, a chemically reactive short-chain alkenal that can covalently modify proteins via 1,2- and 1,4-Michael addition and the Schiff base formation. Generally, 4-HNE is derived from the peroxidation of membrane polyunsaturated fatty acids (PUFAs) such as linoleic, γ-linolenic, or arachidonic acids. The structure of 4-HNE, containing a C2 = C3 double bond, a C1 = O carbonyl group, and a hydroxyl group in the carbon 4 position, makes it a highly reactive aldehyde [14]. Mitochondria, powerhouses of the cell, perform several functions such as ATP generation, calcium homeostasis, and regulation of cell death [15]. Mitochondria are double-membrane organelles with an ample number of lipids. Although most of the lipids are synthesized in the endoplasmic reticulum, de novo lipid synthesis and remodeling of mitochondrial lipids have been shown to play a significant role in maintaining the structural integrity and function of the mitochondria [16]. The most copious phospholipids in the mitochondrial membrane are phosphatidylcholine (PC) and phosphatidylethanolamine (PE), comprising about 40–55% and 30–45% of total mitochondrial phospholipids, respectively. Cardiolipin (CL) accounts for 10–15% of mitochondrial phospholipids. Other common lipids found in the mitochondria in modest numbers are phosphatidylserine (PS), sphingolipids, and sterols [17,18]. A significant role of CL includes the organization of cristae, execution of apoptosis, mitochondrial respiration, mitochondrial dynamics via fission and fusion, carrier protein activity, and signaling during apoptosis [19]. Mitochondria act as a critical source and target of reactive oxygen species [20]. The major source of ROS inside the mitochondria is the respiratory chain complex under physiological and pathological conditions [21]. Due to the high content of unsaturated fatty acid and its location near the site of generation of ROS, CL is highly prone to lipid peroxidation [22]. Cardiolipin peroxidation is linked to several pathological conditions such as ischemia-reperfusion, non-alcoholic fatty liver, hyperthyroidism, traumatic brain injury, and atherosclerosis [23,24,25,26,27]. Recently, it has been shown that the bioactive lipid 4-HNE is also derived from the peroxidation of a mitochondrial-specific phospholipid, CL [27,28,29]. The mechanism of formation of 4-HNE from cardiolipin includes cross-chain peroxyl radical addition and decomposition [29].

## 3. Metabolism of 4-HNE

The metabolism of 4-HNE depends on the location, cell type, and concentration of 4-HNE present on a particular site. Aldehyde dehydrogenases (ALDH), aldose reductase (AR), and glutathione S-transferases (GST) are the major enzymes that metabolize and control 4-HNE levels in cells [30,31,32]. Aldehyde dehydrogenases are major enzymes that play a significant role in the metabolism of 4-HNE and other xenogeneic and biogenic aldehydes [33,34]. 4-HNE is oxidized into a non-toxic 4-hydroxy-2-nonenoic acid by ALDH. Multiple ALDH isozymes in different organs contribute to cytoprotection against 4-HNE and other reactive-aldehyde-induced injuries. ALDH2 is primarily present in mitochondria and is involved in the metabolism of 4-HNE in the heart under disease conditions. It has been shown that 4-HNE inactivates ALDH, and its levels decreased in some pathological hearts [34,35]. The overexpression of ALDH has been found to be protective against 4-HNE, whereas its deletion causes various cardiovascular consequences. The presence of ALDH is significantly beneficial against 4-HNE in metabolic diseases such as cancer and diabetes [36,37,38]. The primary therapeutics targeting 4-HNE are antioxidants such as *N*-acetylcysteine and α-lipoic acid, which enhance ALDH activity and expression [39,40]. It was found that ALDH2 activation can transform deleterious 4-HNE-mediated autophagy into a beneficial state; this effect is mediated by the upregulation of phosphoryl AMPK and Akt during various cardiovascular diseases. The expression and activity of ALDH2 are considered promising therapeutic strategies for reducing the effect of 4-HNE on multiple diseases [41]. In recent days, there has been an attempt to use Alda-1, a molecular-weight activator of ALDH2, which has been shown to attenuate 4-HNE adduction in various pathological conditions such as stroke, hyperoxia-induced lung injury, hypoxia-induced pulmonary hypertension, hepatic ischemia-reperfusion injury, and atherosclerosis [42,43,44,45,46].

Various chains of saturated and unsaturated aldehydes are catalyzed by aldose reductase and metabolize aldehyde–glutathione conjugates such as glutathione S–HNE. AR inhibition leads to 4-HNE accumulation in the heart and blood vessels. It has been reported that 4-HNE can be efficiently detoxified by the aldo-keto reductase aldose reductase, which is purified from the bovine lens [47,48]. A new, efficient detoxification route of HNE is through the enzymatic reaction of aldose reductase, where the natural substrate HNE and HNE-GS are reduced [49]. GST is the most critical determinant of cellular 4-HNE, which detoxifies 4-HNE in most cells. 4-HNE is metabolized when it binds with the GSH, thereby forming a glutathione S conjugate of 4-HNE (GS–HNE) and reducing its toxicity to the cells [50,51]. GSH and GST both play a significant role in the detoxification and protection of cells from 4-HNE-mediated cell injury and damage [50].

## 4. Mitochondrial Dysfunction Mediated by 4-HNE

### 4.1. Exogenous 4-HNE and Mitochondrial Dysfunction

As was mentioned previously, 4-HNE can target protein for its adduction. The mitochondrial protein comprises thirty percent of all the proteins modified by 4-HNE [3]. Both exogenously given and endogenously produced 4-HNE due to oxidative stress have been shown to cause mitochondrial dysfunction in various cell types and organs during physiological and pathological conditions. Small airway epithelial cells (SAECs) form a continuous lining around alveoli and provide a barrier and immunological protection to the lung. These cells are susceptible to damage by oxidative stress, therefore, need to be protected in order to maintain the epithelial membrane barrier to the airway [52,53,54]. A previous study performed in a hyperoxic acute lung injury mice model showed that HNE–protein adducts increase in the lung after exposure to hyperoxia. When human small airway epithelial cells (SAECs) were treated with 25 μM of 4-HNE, it caused a significant decrease in oxygen consumption rate, ATP production, membrane potential, and activity of aconitase [55]. Although they showed that 4-HNE leads to impairment of the mitochondrial energy generation process and a decrease in SAECs survival, they did not perform any study to show the direct modification of any mitochondrial respiratory chain proteins by 4-HNE. Previous studies demonstrated that alternation in mitochondrial dynamic processes, such as fission and fusion, leads to neuronal dysfunction [56,57]. A study performed on primary neurons showed decreased mitochondrial length when they were treated with a 15 μM concentration of 4-HNE. 4-HNE also caused a decrease in complex I and complex V activity, mostly at the 15 μM concentration. 4-HNE treatment also had an effect on mitochondrial fission and fusion proteins. 4-HNE treatment decreased the mitochondrial fusion proteins, such as OPA1 and MFN2, whereas it increased the protein level of fission proteins p-DNM1L and DNM1L, which might have led to a decrease in mitochondrial length [58]. When PC12 cells were treated with 4-HNE, there was a 70% decrease in cytochrome C oxidase activity and a reduction in the activity of mitochondrial matrix enzyme aconitase [59]. Mitochondrial bioenergetic efficiency changes with the level of skeletal muscle activity. When the impact of lipid peroxidation was tested on C2C12 myotubes by directly treating them with 4-HNE, it decreased ATP production, mitochondrial respiration, and spare respiratory capacity. Treatment with 4-HNE also showed an increment in NADH inside the mitochondria indicative of a reduction in complex I activity [60]. However, the direct role of 4-HNE was not examined using a mice model of skeletal muscle injury in this study. Mitochondria are a major source of ROS production; therefore, mitochondrial proteins are highly susceptible to oxidative modification [61]. Previously, the presence of 4-HNE-modified mitochondrial respiratory chain complexes (I–V) was reported in adult bovine heart submitochondrial particles (SMP). When 4-HNE was added to the brain mitochondria, it was shown to significantly increase 4-HNE-bound proteins and decrease mitochondrial complex-I- and complex-II-mediated respiration [62].

### 4.2. Role of 4-HNE Causing Mitochondrial Dysfunction in Various Diseases

Cardiac remodeling after myocardial infarction may lead to heart failure. Oxidative stress is one of the mechanisms by which cardiac remodeling occurs after myocardial infarction [63]. A study performed by Katia et al. in post-cardiac remodeling showed that 4-HNE adducted protein augmented during myocardial ischemia cardiomyopathy progression [64]. The voltage-dependent anion channel (VDAC) and mitochondrial calcium uniporter (MCU) are Ca^2+^ transport channels located in the outer membrane of mitochondria (OMM) and inner membrane of the mitochondrion (IMM), respectively [65]. A study by Saten et al. recently demonstrated that MAO-A augmentation after myocardial ischemia is a source of H_2_O_2_, leading to the generation of reactive aldehyde inside the mitochondria. 4-HNE produced inside the mitochondria is produced via peroxidation of cardiolipin. In the case of post-ischemic cardiac remodeling, CL-peroxidation-derived 4-HNE adducts with the VDAC and MCU lead to the accumulation of calcium inside the mitochondria of cardiomyocytes (Table 1) [66]. This results in increased scar formation and reduced systolic function, as shown by the ejection fraction and fractional shortening. When they used the MAO-A inhibitor moclobemide, they reduced the 4-HNE adduct and improved cardiac function [66]. In patients with congenital heart disease, 4-HNE modified mitochondrial proteins related to metabolisms such as NADH dehydrogenase [ubiquinone] iron–sulfur protein 3, NADH dehydrogenase [ubiquinone] iron–sulfur protein 2, elongation factor Tu, dihydrolipoyl dehydrogenase, ES1 protein homolog, fumarate hydratase, creatine kinase S-type, cytochrome b–c1 complex subunit 1, aconitate hydratase, NADH dehydrogenase [ubiquinone] 1 alpha subcomplex subunit 10 cytochrome c1, heme proteins such as myoglobin, and cytochrome c1, along with other structural and survival-related proteins. 4-HNE also led to a decrease in oxygen consumption by decreasing the activity of mitochondrial respiratory chain complex proteins. All these modifications of proteins by 4-HNE affected the cardiomyocyte energy generation process, leading to the development of right ventricle failure [67].

It has been reported that 4-HNE activates the translocation of the mitochondrial apoptosis-inducing factor (AIFM2) and facilitates apoptosis in the heart tissue of mice and humans. It has also been found that xenobiotics such as doxorubicin significantly enhance HNE levels, enabling an increase in the levels of AIFM2. HNE adduction of AIFM2 inactivates the NADH oxidoreductase activity of AIFM2 and facilitates its translocation from the mitochondria. The main target of HNE adduction that triggers the functional switch is HIS 174 on AIFm2. Recently, we demonstrated that a deficiency in *SOD2* in the conditional-specific cardiac knockout mouse increases ROS, leading to subsequent overproduction of 4-HNE inside mitochondria [72]. Cardiomyocyte-specific *SOD2* knockout mice (*SOD2*^∆^) had shorter life spans and died due to complications of dilated cardiomyopathy. Mechanistically, proteins in the mitochondrial respiratory chain complex and TCA cycle (NDUFS2, SDHA, ATP5B, and DLD) were the target of 4-HNE adduction in *SOD2*^∆^ hearts [72]. Consistent with the results from previous studies, which demonstrated that the respiratory complex proteins NDUFS2 and SDHA are targets of 4-HNE [66,71], we found that both proteins were also modified by 4-HNE in our *SOD2*^∆^ mice with dilated cardiomyopathy [72]. Deficiency in *SOD2* caused an increase in 4-HNE and a reduction in the activity of the mitochondrial complex. Isolated, adult cardiac mitochondria and neonatal cardiomyocytes [73] from *SOD2*^∆^ mice displayed reduced O_2_ consumption, particularly during basal conditions and after the addition of FCCP (H^+^ ionophore/uncoupler), compared to wild-type hearts [72]. This demonstrated the effect of 4-HNE on mitochondrial stress signaling and heart failure and its role in the side effects of various xenobiotic drugs [7]. Therefore, 4-HNE adversely affects various aspects of mitochondrial function, such as oxidative phosphorylation, TCA cycle, calcium homeostasis, mitochondrial fission, and fusion.

It has been established that diabetes is a risk factor for cardiovascular diseases [74,75]. Several studies have shown that oxidative stress is associated with diabetic hearts [76,77]. The characteristics of damaged mitochondria were seen in an animal model of a diabetic heart, where a decrease in respiratory complex I and V activities was seen with an increase in lipid droplets and swelling in mitochondria [78]. Diabetic hearts in the mouse exhibited increased 4-HNE-adducted protein in the mitochondria. In the heart, 4-HNE particularly modified mitochondrial respiratory chain protein succinate dehydrogenase (SDH), and adduction occurred in the FAD-containing subunit of SDH. Mitochondria also showed decreased activity in mitochondrial respiratory chain complex I and complex II activity [68].

Chronic ethanol consumption leads to oxidative stress either by causing ROS generation or decreasing antioxidant concentrations [79]. Ethanol consumption leads to excessive production of mitochondrial ROS, leading to the opening of mitochondrial transition pores and apoptosis [80]. In addition, chronic ethanol consumption causes the accumulation of 4-HNE inside the mitochondria of liver tissue and adducts with mitochondrial HMG-CoA synthase, leading to its inactivation [69]. Chronic ethanol consumption in the rat has been shown to cause an increase in 4-HNE and its adduction with several mitochondrial proteins, including mitofilin, dimethylglycine dehydrogenase, choline dehydrogenase, electron transfer flavoprotein α, cytochrome c1, enoyl CoA hydratase, and cytochrome c, and these protein adductions may contribute to ethanol-mediated mitochondrial dysfunction [81]. It has been reported that oxidative stress leads to the deterioration of aging oocytes. A study on aged oocytes revealed an accumulation of 4-HNE due to increased cytosolic ROS production [82]. Exposure to 4-HNE revealed that 4-HNE causes a decrease in mitochondrial membrane potential and eventual apoptosis of the metaphase II (MII)-stage oocyte. Furthermore, they identified that the accumulation of 4-HNE leads to adduction to the mitochondrial electron transport chain protein SDHA and possibly leads to collapse in the electron transport chain [70].

## 5. Signaling and Cytotoxic Functions of 4-HNE

4-HNE has been established as a signaling molecule, although it is still considered to be a toxic end product of lipid peroxidation. Earlier studies obscured the importance of 4-HNE and how it reacts with the proteins, DNA, and lipids generating the signaling pathway. The various signaling pathways induced by the involvement of 4-HNE include apoptosis, stress response, cell cycle, activation of multiple kinases, gene expression, and the inhibition or activation of critical enzymes [83,84,85,86].

### 5.1. Role of 4-HNE in Nrf2 Signaling

Nrf2 (nuclear factor-E2-related factor 2) is a transcription factor that activates the transcription of a gene through ARE under oxidative stress conditions [87]. Under the homeostatic condition, Nrf2 is subjected to ubiquitination and degradation by its cytosolic inhibitor Keap-1 (Kelch-like ECH-associated protein 1) [88]. When cells are under oxidative stress, oxidation of cysteine residue on Keap-1 causes its conformation change, which prevents binding to the Nrf2, leading to its nuclear translocation [89]. The study showed that 4-HNE protects cardiomyocytes during ischemia-reperfusion injury by activating Nrf2-mediated gene expression and GSH biosynthesis. Lower concentrations of 4-HNE lead to the induction of transcription factor Nrf2 (nuclear factor erythroid-2-related factor 2) by disrupting binding with Keap-1. This increase in Nrf2 causes an increase in the biosynthesis of the antioxidant glutathione (GSH) via increased expression of γ-glutamylcysteine ligase (GCL), a rate-limiting enzyme in glutathione biosynthesis and the core subunit of the Xc high-affinity cystine transporter (xCT), thereby protecting cardiomyocytes during ischemia-reperfusion injury [90] (Figure 2).

### 5.2. Role of 4-HNE in NFkB Signaling

Depending on the concentration, 4-HNE can be beneficial or detrimental to cells. Many studies in the literature suggest that a 4-HNE concentration below 2 μM leads to cell survival and proliferation. However, a concentration above 10 μM is detrimental to the cell, leading to genotoxicity and cell death [91]. 4-HNE at micro-molecular concentrations leads to the upregulation of nuclear factor kappa B (NF-κB), thereby regulating the expression of kinases, such as protein kinase C (PKC) and mitogen-activated protein kinase (MAPK), involved in cellular proliferation and differentiation. On the other hand, increased concentrations of 4-HNE inhibit NF-κB expression [91]. At lower concentrations, 4-HNE binds with IKK and Ik-B, leading to expression of pro-inflammatory gene expression in human fibroblast cells [92] (Figure 2). The study also suggested that 4-HNE regulates apoptosis signaling via the activation of JNK and c-Jun/AP-1 protein [93]. A preponderance of evidence shows that 4-HNE triggers signals that modulate focal adhesion and adherent junction proteins, which induce endothelial barrier dysfunction. As a result of oxidative stress, the overproduction of 4-HNE and oxidized phospholipid (Ox-PL) occurs, contributing to the pathophysiology of various diseases by modulating signaling pathways that regulate pro- and anti-inflammatory responses and barrier regulation [94]. 4-HNE affects the signaling path and endothelial barrier dysfunction by modulating the activities of proteins/enzymes by Michael adduct formation, enhancing the level of protein tyrosine phosphorylation of the target proteins and reorganizing cytoskeletal, focal adhesion, and adherent junction proteins [95]. A study carried out investigating vascular endothelial growth factor (VEGF) in retinal pigment epithelial (RPE) cells showed that a low concentration of 4-HNE (0.1 μM) leads to increased secretion of VEGF, while its expression is blocked when the 4-HNE concentration is greater than 5 μM. This effect of 4-HNE is regulated by GSTA4-4 [96]. Furthermore, 4-HNE was found to cause instability in atherosclerotic plaques by enhancing the release of cytokines, such as IL-8, IL-1β, and TNF-α, thereby causing the upregulation of matrix metalloproteinase-9 (MMP-9) via the TLR4/NF-κB B-dependent pathway [97].

## 6. Role of 4-HNE in Aging

It has been found that the HNE concentration in either free or adduct form increases with aging [92]. 4-HNE also contributes as a signaling molecule in aging. HNE is involved in aging-related signaling pathways through multiple entry points, such as NF-κB, AKT, Nrf2, and mTOR. HNE can control the mTOR signaling network via modulating AKT signaling or through the liver kinase B1 (LKB1) -AMPK-mTORC1 pathway. HNE (40 μM) can conjugate with LKB1, thereby inhibiting AMPK activity and leading to the activation of mTOR/p70S6K–RPS6 signaling (mTORC1) in isolated cardiomyocytes, HEK293T cells, and rat ventricular cardiomyocytes [41,98,99,100]. Furthermore, mTORC1 phosphorylation modulates mitochondria function and autophagy [101,102,103]. Direct evidence of HNE–AMPK1 adducts is yet to be demonstrated, even though HNE inhibits AMPK activity [41,98,99,104]. In addition to these entry points, 4-HNE triggers EGFR related to growth factors and PDGFR [105]. It has been shown that 4-HNE causes skin fibroblast senescence triggered by UV-A exposure. This effect of 4-HNE is contributed by modifying vimentin filaments in cultured skin fibroblasts [106].

## 7. Role of 4-HNE in Stroke

Stroke, one of the major causes of mortality worldwide, still lacks proper markers and interventions for its management; however, oxidative stress markers are being considered for this role [107,108]. 4-HNE, which is produced in response to oxidative insult in the brain following stroke [107], is a major product formed by lipid peroxidation and has been shown to be elevated in stroke patients following acute cerebral ischemia [109]. Lee et al. observed significantly increased levels of plasma 4-HNE in rat models post middle cerebral artery occlusion (MCAO). In addition, 4-HNE was found to be responsible for increased infarct size and oxidative stress. Amplified plasma 4-HNE levels were discovered in a study conducted on patients with post-ischemic stroke [110]. Elevated 4-HNE plasma levels were observed for at least six months following the onset of stroke, and an eight-year follow-up study revealed a relationship between higher initial levels of 4-HNE and the subsequent development of stroke in human subjects [42,111]. It is well known that the cerebral microvascular endothelial cells are essential for the formation of the blood–brain barrier, which is disrupted during ischemic stroke. The genotoxic effect of 4-HNE on these cells, when studied by Karlhuber et al., showed chromosomal aberrations at low concentrations of 4-HNE, whereas higher concentrations showed cytotoxicity with a reduction in mitosis of these cells and increased cell death [112,113].

Aldehyde dehydrogenase 2 family member (ALDH2) is a known mitochondrial enzyme that metabolizes 4-HNE, reducing its toxic effects. In prior studies, it was observed that stroke-prone, spontaneously hypertensive rats (SHR-SP) were deficient in ALDH2, leading to decreased neuroprotection [42]. A study by Li et al. showed reduced levels of 4-HNE following the administration of Alda-1, an ALDH2 agonist, in a rat MCAO model. The increased ALDH2 activity decreased the infarct size and reduced brain edema with an ultimate increase in a neurological deficit. Additionally, they observed a reduction in AQP4 protein expression in the peri-infarct area with ALDH2 activity and reduced edema following an ischemic stroke but increased levels of 4-HNE after MCAO-elevated phosphorylation of ERK1/2 and p38 [114]. PKCε was also shown to regulate ALDH2 in upstream protection against stroke, as ALDH2 knockdown rats demonstrated a loss in neuroprotective effects, but the overexpression of PKCε increased ALDH2 by 44% and decreased infarct by 33% [42]. Together, these data provide evidence supporting the idea that 4-HNE and its regulating molecule ALDH2 can serve as potential biomarkers for stroke.

## 8. Role of 4-HNE in Reproductive Physiology

A recent study suggested that maternally aged oocytes are subjected to oxidative stress burden, leading to increased accumulation of 4-HNE [82,115]. Accumulation of 4-HNE is prominent in oocyte and granulosa cells. 4-HNE decreases the quality of oocytes by causing chromosomal misalignment, meiotic spindle abnormality, microtubule spindle asters formation, and deformities in the microtubule-organizing center (MTOC). The potential target for 4-HNE adduction in the oocyte was found to be α-, β-, and γ-tubulins, leading to increased aneuploidy rates [82]. It has been shown that one of the major causes of defective sperm is oxidative stress. An increase in ROS within spermatozoa leads to the generation of 4-HNE [116]. A study performed on human spermatozoa showed that treatment with an electrophilic aldehyde such as 4-HNE leads to a dose- and time-dependent decrease in the motility of sperm. 4-HNE treatment also leads to an increase in mitochondrial ROS, DNA damage, and apoptosis of human spermatozoa. The potential target for 4-HNE adduction in the mitochondria is SDHA, a subunit of mitochondrial respiratory chain complex II, leading to decreased activity [117].

## 9. Role of Alcohol in 4-HNE-Induced Carcinogenesis

Ethanol and oxidative stress have a xenotoxic effect during carcinogenesis. Ethanol-induced esophageal carcinogenesis occurs via activation of cytochrome P450 2E1 (CYP2E1) and adduct formation with DNA. It has been observed that CYPE1 is highly expressed in patients with esophageal squamous cell carcinoma compared to patients without tumors [118]. In a more recent study, Milloning et al. showed that alcohol increases CYPE1-mediated ROS production in the mucosa of esophageal cancer patients. Furthermore, ROS production subsequently increases the lipid peroxidation products HNE and MDA, which react with DNA bases, resulting in carcinogenic exocyclic–DNA adduct formation in the esophageal mucosa (Figure 3). In addition, 4-HNE causes DNA damage through 2, 3-epoxy-4-hydroxynonenal, and MDA form an exocyclic propane adduct with the N1 and N2 position of guanine (M1dG) [119].

It has been reported that chronic alcohol consumption increases the accumulation of protein adducts with acetaldehyde, MDA, and 4-HNE in the tissues exposed to alcohol. Warnakulasuriya and colleagues showed, for the first time, that immunoreactive CYP2E1 expression is markedly increased in the oral mucosa of squamous cell carcinoma and leukoplakia patients who have a history of chronic alcohol consumption, and that this CYP2E1 expression is closely correlated with acetaldehyde, MDA, and HNE adduct formation in oral keratinocytes [120] (Figure 3).

## 10. Potential Therapeutics Targeting 4-HNE

Growing evidence suggests that mitochondrial aldehyde dehydrogenase detoxifies ROS-mediated aldehyde adducts. Alda-1, the small molecule activator of aldehyde dehydrogenase, has been shown to be beneficial in various pathological conditions such as hypertension [121], stroke [42], chronic heart failure [122], hepatic ischemia/reperfusion injury [45], lung ischemia/reperfusion injury [123], and myocardial ischemia/reperfusion injury [124]. *N*-acetylcysteine (NAC), a thiol reducing agent, possesses antioxidant properties due to a free thiol group [125]. The use of NAC has been shown to be protective against the detrimental effect of 4-HNE, causing protein adduction and depletion of antioxidants [84,126]. The study showed that superoxide dismutase (SOD) mimetic is beneficial in attenuating ROS-induced toxicity in various pathological diseases. As mitochondria are the major site for ROS production, agents targeting the antioxidant defense system, such as superoxide dismutase 2 (MnSOD), can serve as powerful therapeutics. Miriyala et al. demonstrated the potential role of the following MnSOD mimetics to alleviate doxorubicin-induced cardiac injury: MnTnHex-2-PyP^5+^, MnTnBuOE-2-PyP^5+^ (MnP), and MnTE-2-PyP^5+^ [7]. The administration of an exogenous SOD mimetic has the potential to restore the homeostatic balance between oxidants and antioxidants. Specifically, several mitochondrial-based antioxidants are popular in treating heart disease since mitochondria are the major site of ROS production [127]. Different types of metal complexes containing Mn with MnSOD-like activity have been developed over the years. Among them, MnP shows promise as a potential pre-treatment for ROS-induced damage due to its lipophilicity and less toxic profile when compared with other manganese (Mn)-based mimetics [128]. MnP is currently undergoing early-phase clinical trials in reducing oxidative damage caused by radiation and chemotherapy. Furthermore, MnP has been tested and shown to mitigate oxidative damage in several cell types [129,130]. MnP reduces ROS levels either directly or indirectly through inhibition of NF-κB or activation of the Nrf2 signaling pathways [130,131] These mimetics not only restore the oxidant:antioxidant balance, but they also serve as ROS scavengers to help alleviate the ROS-induced injury seen in many pathological diseases. In addition to targeting MnSOD, inhibition of arachidonate 15-lipoxygenase (ALOX15) has been shown to decrease HNE production in human spermatozoa [132]. 4-HNE-induced ROS have been implicated in male infertility because they reduce the function and motility of the sperm [133]. ALOX15 is a lipoxygenase involved in 4-HNE production. Bromfield et al. demonstrated the attenuation of ROS-induced HNE production by pharmacological inhibition with PD146176, a selective ALOX15 inhibitor. Inhibition of ALOX15 decreased mitochondrial ROS as well as cellular ROS, which reduced lipid peroxidation and subsequent 4-HNE formation [6]. Overproduction of ROS in spermatozoa has been linked to infertility; therefore, ALOX15 may serve as another potential therapeutic target in attenuating 4-HNE-induced cell damage. Targeting the enzymes responsible for 4-HNE production, ROS scavengers, and restoring the oxidant:antioxidant balance all serve as potential mechanisms for future therapeutics.

## 11. Conclusions

Mitochondrial cellular bioenergetics and metabolism play a vital role in maintaining cellular homeostasis. 4-HNE formed from the peroxidation of lipids can modify cellular function by covalent adduction with proteins. 4-HNE can be derived from cardiolipin peroxidation by cross-chain peroxyl radical addition and decomposition. 4-HNE can adduct with several mitochondria proteins involved in metabolism, calcium homeostasis, apoptosis, mitochondrial fission, and fusion. The increment of 4-HNE has been linked to the pathogenesis of various diseases, including cardiovascular diseases. ALDH2 activation has been shown to be beneficial in several pathological conditions by causing a decreased level of 4-HNE. Even though most of the evidence cited is from in vitro cell culture studies, new studies are emerging with conditional-specific knockout mouse models for major antioxidant enzymes such as *SOD*2 in the mitochondria [87] to further validate the effect of HNE in an organelle-specific manner.

## Figures and Tables

**Figure 1 biomolecules-12-01555-f001:**
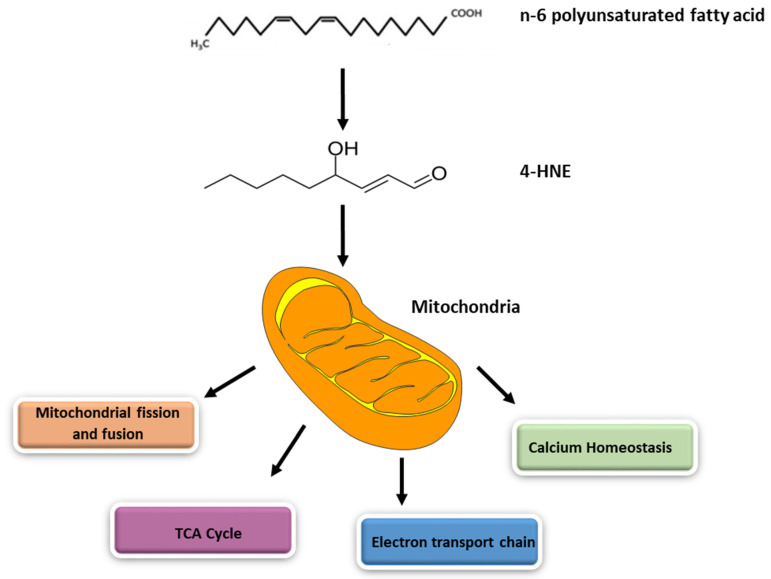
4-HNE generated after peroxidation of n-6 fatty acid affects various aspects of mitochondrial function such as electron transport chain, TCA cycle, mitochondrial fission, mitochondrial fusion, and calcium homeostasis.

**Figure 2 biomolecules-12-01555-f002:**
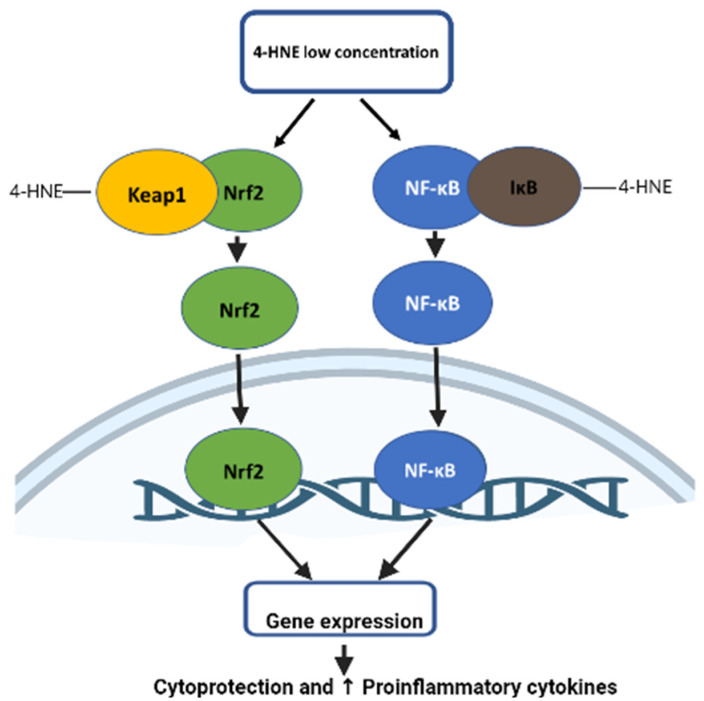
Signaling of 4-HNE in Nrf2 and NF-κB pathway. Lower concentrations of 4-HNE lead to the induction of transcription factor Nrf2 (nuclear factor erythroid-2-related factor 2) and NF-κB (nuclear factor kappa B (NF-κB). This increase in Nrf2 causes an increase in gene expression, which leads to cardioprotection by enhancing synthesis of antioxidants. Increase in NF-κB leads to increased (↑) pro-inflammatory cytokine expression.

**Figure 3 biomolecules-12-01555-f003:**
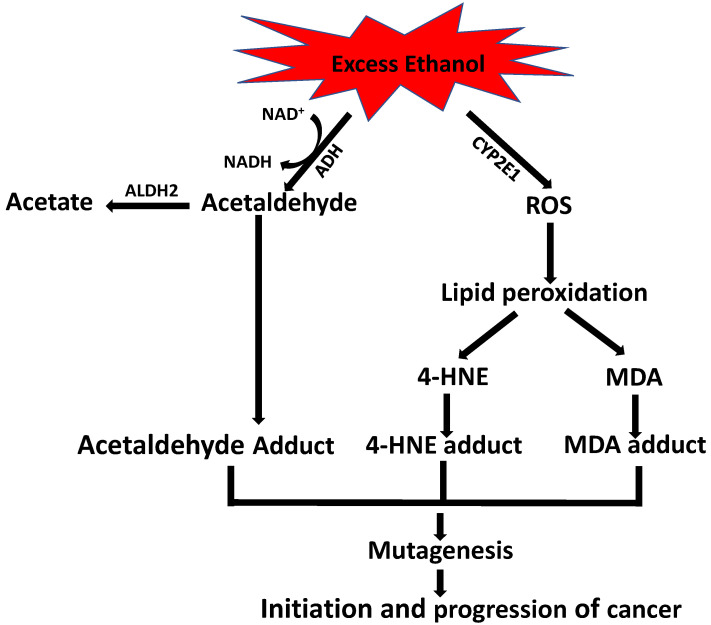
Schematic representation of 4-HNE-induced carcinogenesis due to chronic alcoholism. Chronic/high levels of alcohol consumption cause significant changes in alcohol metabolism by additional activation of CYP2E1, which metabolizes alcohol and produces acetaldehyde as well as ROS. ROS increase lipid peroxidation, leading to the production of the toxic end products 4-HNE and MDA. Increased levels of acetaldehyde, 4-HNE, and MDA react with DNA bases, leading to the production of the carcinogenic adduct, which, subsequently, causes mutation of certain genes, resulting in the initiation and progression of cancer.

**Table 1 biomolecules-12-01555-t001:** Major studies identifying the role of 4-HNE in different mitochondrial functions.

Cell or Tissue	Mitochondria Affected	Adducted Proteins, Mitochondria	References
Myocardial tissue	Calcium accumulation inside mitochondria	VDAC and MCU	Santin, 2020 [66]
Myocardial tissue	Decrease in mitochondrial respiratory chain complex I and II activity	SDH	Lashin, 2006 [68]
Liver tissue	HMG-CoA synthase inactivation	HMG-CoA synthase	Patel, 2007 [69]
RV myocardial tissue	Decreased oxidative phosphorylation	NADH dehydrogenase [ubiquinone] iron–sulfur protein 2, elongation factor Tu, dihydrolipoyl dehydrogenase, ES1 protein homolog, fumarate hydratase, creatine kinase S-type, cytochrome b–c1 complex subunit 1, aconitate hydratase, NADH dehydrogenase [ubiquinone] 1 alpha subcomplex subunit 10, cytochrome c1, heme protein, stress-70 protein, superoxide dismutase	Hwang, 2020 [67]
Murine MII-stage oocyte	Loss of mitochondrial membrane potential	SDHA	Lord, 2015 [70]
Cardiac tissue	Inactivates the oxidoreductase activity of AIFm2	AIFm2	Miriyala, 2017 [7]
Cardiac tissue	Decreased oxygen consumption rate	NADH ATP synthase subunit, dihydrolipoyl dehydrogenase, succinate dehydrogenase [ubiquinone] flavoprotein subunit, trifunctional enzyme subunit α, creatine kinase S-type, cytoplasmic isoform of fumarate hydratase, succinyl-CoA:3-ketoacid–coenzyme A transferase 1	Zhao, 2014 [71]
Myocardial tissue	Decreased oxygen consumption rate, decreased complex I and complex V activity	NDUFS2, SDHA, ATP5B, and DLD	Sharma, 2020 [72]

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
