# Peer review of "Electrophilic Aldehyde 4-Hydroxy-2-Nonenal Mediated Signaling and Mitochondrial Dysfunction"

_biomolecules, 2022, doi:10.3390/biom12111555_

Round 1

Reviewer 1 Report

The role of 4-hydroxynonenal (4-HNE) in mitochondria and cells is important and worth reviewing. Here the authors cover the extensive literature on this topic. Although, overall, this is an interesting and well-written review, I have two major suggestions.1) In some parts of the review (for example, “part 4.1. Role of 4-HNE causing mitochondrial dysfunction in various diseases”, lines 165-185, this is just an example) the authors just listed the results previously published by others, and reading this part is a bit boring. On the contrary, the description of the results obtained by the authors of the submitted manuscript is really well written. A good review does not just summarize the literature, but discusses it critically, identifies problems, and points out research gaps. 2) My second suggestion is to extend the subsection “4.1. Exogenous 4-HNE and mitochondrial dysfunction”. I believe that this part is directly related to the modification of mitochondrial electron transfer complexes by HNE. As far as I know, interesting articles concerning modification of individual ETC by HNE have been published, but are not discussed. Why both subsections are labeled as 4.1.? The same 5.1 and 5.1? t is logical to expect that the readers will see 4.1; 4.2…I have also a few minor specific comments which I have in attached pdf file.
Some of the minor points that may need corrections are marked yellow right in the text. 

Author Response

Enclosed as PDF file

Reviewer 2 Report

In their review manuscript, the authors have comprehensively covered the various effects that the 4-hydroxy-2-nonenal (4-HNE) byproduct of oxidative metabolism has on human mitochondrial function and signal transduction. After a brief outline of the 4-HNE biology, the authors went first on to summarize the mechanisms of how 4-HNE leads to mitochondrial dysfunction in the context of pathophysiology. To this end, they compiled the results of this section into a neat table containing all the relevant references. Next, the authors touched on the crosstalk between 4-HNE and the Nrf2 and NF-κB signaling pathways. Importantly, they finished the coherent story by addressing the ramifications of 4-HNE accumulation in various clinical settings while highlighting potential therapeutic mitigation strategies. The paper is well structured and provides an insightful introspective into the up-to-date knowledge on the 4-HNE problematic. Despite these assets, poor presentation of figures and their correctness constitutes a major hurdle that hinders a complete understanding of the developed concepts (major points, see below). Furthermore, a more detailed attention needs to be paid to grammar usage/style as well (please see the minor points below).

Some points:

1) Despite that the authors discuss the effects of 4-HNE on Nrf2 and NF-κB signaling in sufficient detail, only the Nrf2 pathway is depicted in Figure 2. Please generalize this figure to a greater extent and/or provide a new figure on the specific alterations of NF-κB signaling by 4-HNE.

2) The quality of Figure 2 design is rather meager. Please redraw by using uniform formatting of the arrow symbols, precise horizontal alignment of both the arrow symbols and blue and yellow diagrams. In addition, center the text both horizontally and vertically within each diagram. It might also be a good idea to add a little bit of green color into the figure, perhaps replacing the light-yellow hue used for Nrf2.

3) The quality of Figure 3 is poor as well. Please align all arrow symbols so that the text is exactly centered within them. Please center all enzyme names above their respective blue reaction arrows. Please connect the blue reaction arrows between ethanol and acetaldehyde so that they form a single reaction unit. Please also redraw the yellow star diagram so that the "Excess ethanol" caption does not collide with its border.

4) The ADH reaction presented in Figure 3 is wrong as ethanol is proposed to be reduced to acetaldehyde by NADH according to the reaction scheme. Please revise.

5) The appearance of Figure 3 is rather pixelated. Please increase its size and/or pixel density.

6)  It is not clear what the authors mean by "assigning HNE and HNE-GS to be the best natural substrate of aldose reductase" and "known to do far" in "It also stated that the enzyme displays a Km of congruent to 9 μM for HNE and 34 μM for the glutathione adduct of HNE (HNE-GS), assigning HNE and HNE-GS to be the best natural substrate of aldose reductase known to do far and exposing a new efficient detoxification route of HNE" (line 131)?

7)  It is not exactly clear what the authors mean by "reduced GSH" in "The reduction of toxicity depends on the reaction of 4-HNE with reduced GSH, which forms a glutathione S conjugate of 4-HNE (GS-137 HNE)" (line 136)?

8) Please specify the type of "heme protein" mentioned in "In patients with congenital heart disease, 4-HNE modified mitochondrial proteins related to metabolism such as NADH dehydrogenase [ubiquinone] iron- sulfur protein 3, NADH dehydrogenase [ubiquinone] iron-sulfur protein 2, Elongation factor Tu, Dihydrolipoyl dehydrogenase, ES1 protein homolog, Fumarate hydratase, Creatine kinase S-type, Cytochrome b-c1 complex subunit 1, Aconitate hydratase, and NADH dehydrogenase [ubiquinone] 1 alpha subcomplex subunit 10 Cytochrome c1, heme protein along with other structural and survival-related proteins" (line 177).

Author Response

Enclosed as PDF file

Round 2

Reviewer 1 Report

1. The authors stated that they fixed duplicate subsections with numbers 4.1 and 5.1. But I see the same problem in the revised version. I have no problem if this is a journal policy for manuscript formatting. However, it seems confusing to me.

2. Please, clarify Line 151: "...a 70% decrease in cytochrome C oxidase...". Activity? Amount?

Author Response

Enclosed
